# Design and Synthesis of *N*-phenyl Phthalimides as Potent Protoporphyrinogen Oxidase Inhibitors

**DOI:** 10.3390/molecules24234363

**Published:** 2019-11-29

**Authors:** Wei Gao, Xiaotian Li, Da Ren, Susu Sun, Jingqian Huo, Yanen Wang, Lai Chen, Jinlin Zhang

**Affiliations:** Plant Protection College, Hebei Agricultural University, Baoding 071000, China; hbaugaow@163.com (W.G.); kddanongyao@163.com (X.L.); 18233470747@163.com (D.R.); susu8023d@163.com (S.S.); huojingqian@163.com (J.H.); yanenwang@163.com (Y.W.)

**Keywords:** protoporphyrinogen oxidase, molecular design, *N*-phenyl phthalimides, herbicidal activity

## Abstract

Protoporphyrinogen oxidase (PPO) has been identified as one of the most promising targets for herbicide discovery. A series of novel phthalimide derivatives were designed by molecular docking studies targeting the crystal structure of mitochondrial PPO from tobacco (*mt*PPO, PDB: 1SEZ) by using Flumioxazin as a lead, after which the derivatives were synthesized and characterized, and their herbicidal activities were subsequently evaluated. The herbicidal bioassay results showed that compounds such as **3a** (2-(4-bromo-2,6-difluorophenyl) isoindoline-1,3-dione), **3d** (methyl 2-(4-chloro-1,3-dioxoisoindolin-2-yl)-5-fluorobenzoate), **3g** (4-chloro-2-(5-methylisoxazol-3-yl) isoindoline-1,3-dione), **3j** (4-chloro-2-(thiophen-2-ylmethyl) isoindoline-1,3-dione) and **3r** (2-(4-bromo-2,6-difluorophenyl)-4-fluoroisoindoline-1,3-dione) had good herbicidal activities; among them, **3a** showed excellent herbicidal efficacy against *A. retroflexus* and *B. campestris* via the small cup method and via pre-emergence and post-emergence spray treatments. The efficacy was comparable to that of the commercial herbicides Flumioxazin, Atrazine, and Chlortoluron. Further, the enzyme activity assay results suggest that the mode of action of compound **3a** involves the inhibition of the PPO enzyme, and **3a** showed better inhibitory activity against PPO than did Flumioxazin. These results indicate that our molecular design strategy contributes to the development of novel promising PPO inhibitors.

## 1. Introduction

Protoporphyrinogen oxidase (PPO) is the last common enzyme in both chlorophyll (in plants) and haem (in animals) biosynthesis [1,2,3,4], catalyzing the oxidation of protoporphyrinogen IX to protoporphyrin IX via molecular oxygen [5,6,7,8,9], and this enzyme has been identified as one of the most significant targets for herbicide research [10,11]. During the last thirty years, a number of active compounds inhibiting the enzyme PPO have been synthesized [12,13], some of which have been developed for use as low-toxicity, efficient, broad-spectrum commercial herbicides [14,15], such as Flumioxazin [16,17], sulfentrazone [18] and saflufenacil [19].

Among the protox herbicides, *N*-phenyl phthalimides, which exhibit broad structural diversity [20], have attracted considerable attention; their representative commercial products, Cinidon-ethyl, Flumiclorac-pentyl and Flumioxazin, were identified as a result of a stepwise optimization procedure from Chlorphthalim. Their common structural feature consists of two parts: An *N*-substituted phenyl group and tetrahydrophthalimide, which can interact with key active centre residues of the *mt*PPO enzyme, such as Arg98, Gly175, Leu372, Phe392, and FAD600, via H-bonding or π–π stacking interactions [21]. These findings have contributed to the development of PPO inhibitors.

To obtain novel PPO inhibitors, a series of *N*-phenyl phthalimides were designed by molecular docking using the *mt*PPO as a target, and by using Flumioxazin as a lead (Figure 1) [22,23], after which they were synthesized and characterized by NMR and High resolution mass spectrometry (HR-MS). Their herbicidal activities were also evaluated against *Brassica campestris* (*B. campestris*), *Amaranthus retroflexus* (*A. retroflexus*) and *Digitaria sanguinalis* (*D. sanguinalis*) to verify our molecular design strategy.

## 2. Results and Discussion

### 2.1. Docking Analysis

The affinity values of molecular docking between the ligand-*mt*PPO complexes were determined, which demonstrated that the affinity between **3a**, **3c** and **3s** and *mt*PPO exhibited the highest values with −10.0, −10.1, 10.2 Kcal/mol (Table 1), respectively, suggesting that these compounds may have good herbicidal activity. Their docking models were displayed by Pymol software (Figure 2), which indicated that no hydrogen bonding was found between these compounds and *mt*PPO, which differs from the results of Flumioxazin (three hydrogen bonds). However, the phthalimide rings of compounds **3a**, **3c** and **3s** shape π-π stacking interactions with Phe392, which is conserved in plant PPO enzymes. The phenyl rings of compounds **3a**, **3c** and **3s** were sandwiched by the residues Leu356 and Leu372. These results suggest that these compounds, with a phenyl ring substituting phthalimide rings, could be PPO inhibitors that exhibit good herbicidal activity.

### 2.2. Chemistry

The starting material **1** and **2** could be commercially available. Compounds **3** were prepared by nucleophilic substitution reaction between phthalic anhydrides **1** and amines **2** in glacial acetic acid, with yields ranging from 28% to 87% (Table 2). The reaction solvent and temperature were the key conditions for the reaction yield. The yield of the reaction stirred at 110 °C was much better than that of 80 °C. Besides, the yield of the reaction in glacial acetic acid was higher than that of other solvents, such as ethanol.

### 2.3. Herbicidal Activity

The herbicidal activity against *B. campestris*, *A. retroflexus* and *D. sanguinalis* at 200 mg/L of compound **3** were evaluated by the small cup method, and the results are shown in Table 3. The data showed that **3a** at 200 mg/L displayed 92% growth inhibition against *B. campestris* roots, which was better than that of the positive controls Chlortoluron (85%), Atrazine (80%), and Flumioxazin (85%); **3a** also showed 61% growth inhibition against stems of *B. campestris*, which was better than that of Atrazine (51%) and similar to that of Chlortoluron (58%). During the test, the *B. campestris* plants treated with **3a** became chlorotic, and blade yellowing was also observed. In addition, with 87% growth inhibition, **3a** exhibited better efficacy against *A. retroflexus* stems than did Atrazine, which was similar to that of Flumioxazin (88%); the leaves of *A. retroflexus* were wrinkled. Furthermore, **3a** displayed 68% and 83% growth inhibition against the roots and stems of *D. sanguinalis*, respectively, which were better than the inhibition due to Atrazine. Compounds **3d** and **3o** exhibited high activity against the roots of *A. retroflexus*, with 81% and 83% growth inhibition, respectively, while Atrazine showed only 32% inhibition under the same conditions; **3o** also showed 89% growth inhibition activity against *B. campestris* roots, which was slightly better than that of the three positive controls. **3g** showed excellent efficacy against *D. sanguinalis* roots and stems, with 91% and 83% growth inhibition, respectively, these percentages were better than those due to Atrazine.

The results of the evaluation of the post-emergence herbicidal activities are shown in Table 4. The data showed that compounds **3a** and **3d** exhibited 82% and 73% fresh weight growth inhibition at 90 g ai/ha against *A. retroflexus*, respectively. Therefore, **3a** and **3d** were also chosen for further pre-emergence herbicidal activity tests. As shown in Table 5, **3a** at 90 g ai/ha exhibited a 98% inhibitory effect against *A. retroflexus*, and the effects were not significantly different from those of Flumioxazin.

In summary, **3a** exhibited excellent herbicidal activity and should be further developed. Moreover, its herbicidal activity conformed to the predictions of the molecular docking studies.

### 2.4. Crystal Structure Determination of Compound 3a

The X-ray diffraction structure of compound **3a**, cultured from the mixture of ethanol and chloroform, was shown in Figure 3, and the data were available at the Cambridge Crystallographic Data Centre (CCDC 1923372).

### 2.5. PPO Enzyme Assays

The PPO enzyme activity of *D. sanguinalis,* treated with **3a** and Flumioxazin at 90 g ai/ha, were measured, and the control was sprayed with blank solution without any compounds. The results, shown in Figure 4, indicated that the PPO activity values of the **3a** treatment and the control Flumioxazin were affected; in addition, **3a** showed a stronger effect on the PPO enzyme, with 33.01%, compared to that of the positive control Flumioxazin, with 21.80%.

## 3. Materials and Methods

### 3.1. Molecular Docking

The structure of *mt*PPO was available at the National Center for Biotechnology Information (NCBI) database. The modelled complexes of *mt*PPO and ligands (Flumioxazin, Chlorotoluron, Atrazine and designed target compounds) were prepared by using AutoDockTools 1.5.6 (Molecular Graphics Laboratory, La Jolla, CA 92037-1000, USA) and analysed via the AutoDock Vina program (Molecular Graphics Laboratory, La Jolla, CA 92037-1000, USA) [24,25]. The structures of the complexes were generated by the Pymol tool 2.2.0 [26,27].

### 3.2. Equipment and Materials

The melting points of the new compounds were measured in a microfusion melting point apparatus (X-4) (Taike, Beijing, China) and uncorrected. ^1^H-NMR and ^13^C-NMR spectra were recorded on Varian 400 spectrometer at 400 MHz and 101 MHz using tetramethylsilane as internal standard (solvent CDCl_3_ or DMSO-*d_6_*). HR-MS date were detected on an FTICR-MS Varian 7.0T FTICR-MS equipment (Agilent, Lexington, MA, USA). Crystal structure was recorded on a Bruker SMART 1000CCD diffraction meter. 

### 3.3. General Synthetic Procedure for Compounds 3

The starting materials **1** and **2** were commercially available (Energy Chemical, Shanghai, China). Compound **2** (3.72 mmol) was added to a stirred solution of compound **1** (3.38 mmol) in glacial acetic acid (10 mL). The reaction mixture was then stirred at 110 °C for 4 h. After completion of the reaction, the solvent was evaporated, and the residue was purified on a silica gel column chromatography and eluted with ethyl acetate/petroleum ether (bp 60–90 °C) (1:3, *v*/*v*) to give compounds **3**. Among them, compounds **3k**, **3l**, **3n**, **3x** and **3y** were published [28,29,30,31,32], while their herbicidal activities were not studied. Their yields, physical properties, ^1^H-NMR, ^13^C-NMR, and HR-MS results are shown as follows:

Data for **3a** (2-(4-bromo-2,6-difluorophenyl)isoindoline-1,3-dione): white solid; yield, 73%; m.p.: 175–176 °C; ^1^H-NMR (400 MHz, CDCl_3_) δ 7.98 (dd, *J* = 5.5, 3.1 Hz, 2H), 7.83 (dd, *J* = 5.5, 3.1 Hz, 2H), 7.29 (d, *J* = 6.7 Hz, 2H).^13^C-NMR (101 MHz, CDCl_3_) δ 165.41 (s), 160.09 (s), 157.47 (s), 134.73 (s), 131.93 (s), 124.22 (s), 123.29 (s), 116.52 (s), 116.25 (s). HR-MS (ESI) [M + H]^+^ calcd for C_14_H_6_BrF_2_NO_2_: 337.9500, found: 337.9621.

Data for **3b** (5-chloro-2-(5-methylisoxazol-3-yl)isoindoline-1,3-dione): white solid; yield, 77%; m.p.: 164–165 °C; ^1^H-NMR (400 MHz, CDCl_3_) δ 7.95 (dd, *J* = 12.4, 4.8 Hz, 2H), 7.79 (dd, *J* = 8.0, 1.7 Hz, 1H), 6.48 (s, 1H), 2.51 (s, 3H). ^13^C-NMR (101 MHz, CDCl_3_) δ 170.98 (s), 163.90 (s), 163.65 (s), 153.28 (s), 141.86 (s), 135.13 (s), 133.13 (s), 129.55 (s), 125.56 (s), 124.67 (s), 97.96 (s), 12.73 (s). HR-MS (ESI) [M + H]^+^ calcd for C_12_H_7_N_2_O_3_: 263.0145, found: 263.0223.

Data for **3c** (4-chloro-2-(3-fluoro-4-methylphenyl)isoindoline-1,3-dione): white solid; yield, 81%; m.p.: 197–198 °C; ^1^H-NMR (400 MHz, CDCl_3_) δ 7.95–7.80 (m, 1H), 7.79–7.60 (m, 2H), 7.30 (d, *J* = 8.1 Hz, 1H), 7.14 (t, *J* = 8.3 Hz, 2H), 2.32 (d, *J* = 7.1 Hz, 3H). ^13^C-NMR (101 MHz, CDCl_3_) δ 165.46 (s), 162.00 (s), 159.55 (s), 136.11 (s), 135.23 (s), 133.58 (s), 131.88 (s), 131.49 (s), 129.90 (s), 127.14 (s), 125.23 (s), 122.18 (s), 121.74 (s), 113.59 (s), 14.27 (s). HR-MS (ESI) [M + H]^+^ calcd for C_15_H_9_ClFNO_2_: 290.0306, found: 290.0380. 

Data for **3d** (methyl 2-(4-chloro-1,3-dioxoisoindolin-2-yl)-5-fluorobenzoate): white solid; yield, 28%; m.p.: 136–137 °C; ^1^H-NMR (400 MHz, CDCl_3_) δ 7.89 (d, *J* = 3.8 Hz, 2H), 7.74 (d, *J* = 3.8 Hz, 2H), 7.40 (d, *J* = 4.6 Hz, 2H), 3.80 (s, 3H). ^13^C-NMR (101 MHz, CDCl_3_) δ 165.96 (s), 164.93 (s), 163.85 (s), 160.96 (s), 136.04 (s), 135.19 (s), 134.03 (s), 132.32 (s), 131.86 (s), 129.68 (s), 127.72 (s), 127.45 (s), 122.25 (s), 120.48 (s), 118.88 (s), 52.62 (s). HR-MS (ESI) [M + H]^+^ calcd for C_16_H_9_ClFNO_4_: 334.0204, found: 334.0273. 

Data for **3e** (2-(4-bromo-2,6-difluorophenyl)-4-chloroisoindoline-1,3-dione): white solid; yield, 80%; m.p.: 185–186 °C; ^1^H-NMR (400 MHz, CDCl_3_) δ 7.94–7.82 (m, 2H), 7.74 (d, *J* = 3.8 Hz, 2H), 7.45–7.35 (m, 2H), 3.80 (s, 3H).^13^C-NMR (101 MHz, CDCl_3_) δ 163.97 (s), 162.87 (s), 159.98 (s), 157.41 (s), 136.47 (s), 135.60 (s), 133.91 (s), 132.39 (s), 127.72 (s), 123.57 (s), 122.67 (s), 116.55 (s), 116.28 (s), 108.22 (s). HR-MS (ESI) [M + H]^+^ calcd for C_14_H_5_BrClF_2_NO_2_: 371.9160, found: 371.9232.

Data for **3f** (4-chloro-2-(4-isopropylphenyl)isoindoline-1,3-dione): white solid; yield, 78%; m.p.: 154–155 °C; ^1^H-NMR (400 MHz, CDCl_3_) δ 7.89 (dd, *J* = 4.4, 3.9 Hz, 1H), 7.76–7.68 (m, 2H), 7.43–7.32 (m, 4H), 2.99 (s, 1H), 1.31 (d, *J* = 6.9 Hz, 6H). ^13^C-NMR (101 MHz, CDCl_3_) δ 166.04 (s), 165.03 (s), 149.13 (s), 136.09 (s), 135.22 (s), 133.93 (s), 131.85 (s), 128.83 (s), 127.46 (s), 127.25 (s), 126.42 (s), 122.21 (s), 33.95 (s), 23.93 (s). HR-MS (ESI) [M + H]^+^ calcd for C_17_H_14_ClNO_2_: 300.0713, found: 300.0791. 

Data for **3g** (4-chloro-2-(5-methylisoxazol-3-yl)isoindoline-1,3-dione): White solid; yield, 58%; m.p.: 160–161 °C; ^1^H-NMR (400 MHz, CDCl_3_) δ 7.95–7.89 (m, 1H), 7.76 (d, *J* = 4.4 Hz, 2H), 6.49 (s, 1H), 2.52 (s, 3H). ^13^C-NMR (101 MHz, CDCl_3_) δ 170.96 (s), 163.34 (s), 162.52 (s), 153.18 (s), 136.69 (s), 135.83 (s), 133.59 (s), 132.52 (s), 127.21 (s), 122.81 (s), 98.09 (s), 12.76 (s). HR-MS (ESI) [M + H]^+^ calcd for C_12_H_7_ClN_2_O_3_: 263.0145, found: 263.0220. 

Data for **3h** (4-chloro-2-(1-methyl-1H-pyrazol-5-yl)isoindoline-1,3-dione): white solid; yield, 60%; m.p.: 195–196 °C; ^1^H-NMR (400 MHz, CDCl_3_) δ 7.91 (dd, *J* = 4.6, 3.7 Hz, 1H), 7.81–7.74 (m, 2H), 7.60 (d, *J* = 2.0 Hz, 1H), 6.33 (d, *J* = 2.0 Hz, 1H), 3.78 (s, 3H).^13^C-NMR (101 MHz, CDCl_3_) δ 164.65 (s), 163.59 (s), 138.84 (s), 136.69 (s), 135.83 (s), 133.61 (s), 132.52 (s), 129.13 (s), 127.39 (s), 122.76 (s), 104.67 (s), 36.51 (s). HR-MS (ESI) [M + H]^+^ calcd for C_12_H_8_ClN_3_O_2_: 262.0305, found: 262.0379. 

Data for **3i** (4-chloro-2-(3,4-dichlorobenzyl)isoindoline-1,3-dione): white solid; yield, 67%; m.p.: 143–144 °C; ^1^H-NMR (400 MHz, CDCl_3_) δ 7.78 (dd, *J* = 4.8, 3.4 Hz, 1H), 7.69–7.62 (m, 2H), 7.53 (d, *J* = 2.0 Hz, 1H), 7.39 (d, *J* = 8.2 Hz, 1H), 7.30 (d, *J* = 2.0 Hz, 1H), 4.78 (s, 2H). ^13^C-NMR (101 MHz, CDCl_3_) δ 166.32 (s), 165.47 (s), 135.99 (d, *J* = 4.7 Hz), 135.18 (s), 133.97 (s), 132.77 (s), 132.30 (s), 131.65 (s), 130.76 (d, *J* = 8.8 Hz), 128.30 (s), 127.63 (s), 122.05 (s), 40.70 (s). HR-MS (ESI) [M + H]^+^ calcd for C_15_H_8_Cl_3_NO_2_: 339.9621, found: 339.9692.

Data for **3j** (4-chloro-2-(thiophen-2-ylmethyl)isoindoline-1,3-dione): white solid; yield, 71%; m.p.: 135–136 °C; ^1^H-NMR (400 MHz, CDCl_3_) δ 7.78 (dd, *J* = 4.6, 3.6 Hz, 1H), 7.68–7.61 (m, 2H), 7.31–7.15 (m, 2H), 6.95 (dd, *J* = 5.1, 3.6 Hz, 1H), 5.03 (s, 2H).^13^C-NMR (101 MHz, CDCl_3_) δ 166.09 (s), 165.20 (s), 137.56 (s), 135.81 (s), 135.03 (s), 134.09 (s), 131.54 (s), 128.08 (s), 127.74 (s), 126.95 (s), 126.08 (s), 121.95 (s), 35.82 (s). HR-MS (ESI) [M + H]^+^ calcd for C_13_H_8_ClNO_2_S: 277.9964, found: 278.0038.

Data for **3k** (4-chloro-2-(3-fluoro-4-nitrophenyl)isoindoline-1,3-dione): brown solid, yield, 63%; m.p.: 231–232 °C; ^1^H-NMR (400 MHz, DMSO) δ 8.42 (dd, *J* = 9.7, 2.1 Hz, 1H), 8.31 (d, *J* = 8.7 Hz, 1H), 7.95 (ddd, *J* = 26.6, 16.0, 7.9 Hz, 4H).^13^C-NMR (101 MHz, DMSO) δ 164.72 (s), 163.70 (s), 158.45 (s), 155.91 (s), 148.65 (s), 137.06 (s), 134.29 (s), 131.84 (s), 130.71 (s), 127.86 (s), 125.75 (s), 123.36 (s), 120.71 (s), 113.21 (s). HR-MS (ESI) [M + H]^+^ calcd for C_14_H_6_ClFN_2_O_4_: 321.0000, found: 321.0059.

Data for **3l** (4-chloro-2-(4-phenoxyphenyl)isoindoline-1,3-dione): white solid; yield, 86%; m.p.: 172–173 °C; ^1^H-NMR (400 MHz, CDCl_3_) δ 8.05–7.98 (m, 1H), 7.86 (d, *J* = 4.3 Hz, 2H), 7.52 (dd, *J* = 4.7, 4.0 Hz, 4H), 7.35–7.14 (m, 5H).^13^C-NMR (101 MHz, CDCl_3_) δ 165.97 (s), 164.98 (s), 157.39 (s), 156.44 (s), 136.16 (s), 135.29 (s), 133.86 (s), 131.92 (s), 129.94 (s), 128.07 (s), 127.40 (s), 125.97 (s), 123.97 (s), 122.25 (s), 119.60 (s), 118.81 (s). HR-MS (ESI) [M + H]^+^ calcd for C_20_H_12_ClNO_3_: 350.0506, found: 350.0580.

Data for **3m** (3-(4-chloro-1,3-dioxoisoindolin-2-yl)benzamide): light yellow solid; yield, 84%; m.p.: 252–253 °C; ^1^H-NMR (400 MHz, DMSO) δ 8.09 (s, 1H), 8.03–7.83 (m, 5H), 7.62 (s, 2H), 7.53 (s, 1H). ^13^C-NMR (101 MHz, CDCl_3_) δ 172.19 (s), 170.82 (s), 169.85 (s), 141.41 (s), 141.12 (s), 140.34 (s), 139.16 (s), 136.99 (s), 135.44 (s), 135.05 (s), 134.05 (s), 132.53 (s), 132.23 (s), 132.10 (s), 127.60 (s). HR-MS (ESI) [M + H]^+^ calcd for C_15_H_9_ClN_2_O_3_: 301.0302, found: 301.0380.

Data for **3n** (2-(4-(tert-butyl)phenyl)-4-chloroisoindoline-1,3-dione): white solid; yield, 52%; m.p.: 178–179 °C; ^1^H-NMR (400 MHz, CDCl_3_) δ 7.93–7.82 (m, 1H), 7.70 (d, *J* = 3.8 Hz, 2H), 7.52 (d, *J* = 8.5 Hz, 2H), 7.35 (d, *J* = 8.4 Hz, 2H), 1.36 (s, 9H).^13^C-NMR (101 MHz, CDCl_3_) δ 166.02 (s), 165.01 (s), 151.35 (s), 136.08 (s), 135.21 (s), 133.94 (s), 131.84 (s), 128.59 (s), 127.47 (s), 126.18 (s), 126.04 (s), 122.21 (s), 34.76 (s), 31.32 (s). HR-MS (ESI) [M + H]^+^ calcd for C_18_H_16_ClNO_2_: 314.0870, found: 314.0947.

Data for **3o** (4-chloro-2-(3-fluoro-2-methoxyphenyl)isoindoline-1,3-dione): white solid; yield, 87%; m.p.: 153–154 °C; ^1^H-NMR (400 MHz, CDCl_3_) δ 8.10–7.99 (m, 1H), 7.97–7.81 (m, 2H), 7.46–7.37 (m, 1H), 7.30 (td, *J* = 8.1, 5.2 Hz, 1H), 7.26–7.15 (m, 1H), 4.14 (d, *J* = 2.4 Hz, 3H).^13^C-NMR (101 MHz, CDCl_3_) δ 165.63 (s), 164.59 (s), 156.89 (s), 154.43 (s), 144.39 (s), 136.11 (s), 135.25 (s), 134.10 (s), 131.93 (s), 127.76 (s), 125.01 (s), 123.03 (s), 122.32 (s), 118.40 (s), 61.42 (s). HR-MS (ESI) [M + H]^+^ calcd for C_15_H_9_ClFNO_3_: 306.0255, found: 306.0331.

Data for **3p** (4-chloro-2-(5-methylthiazol-2-yl)isoindoline-1,3-dione): yellow solid; yield, 30%; m.p.: 143–144 °C; ^1^H-NMR (400 MHz, CDCl_3_) δ 7.83 (dd, *J* = 63.0, 3.8 Hz, 3H), 6.93 (s, 1H), 2.52 (s, 3H).^13^C-NMR (101 MHz, CDCl_3_) δ 163.32 (s), 162.62 (s), 150.64 (s), 150.45 (s), 136.77 (s), 135.92 (s), 133.30 (s), 132.62 (s), 126.87 (s), 122.81 (s), 113.01 (s), 17.48 (s). HR-MS (ESI) [M + H]^+^ calcd for C_12_H_7_ClN_2_O_2_S: 278.9917, found: 278.9988.

Data for **3q** (methyl 5-fluoro-2-(4-fluoro-1,3-dioxoisoindolin-2-yl)benzoate): white solid; yield, 57%; m.p.: 179–180 °C; ^1^H-NMR (400 MHz, CDCl_3_) δ 8.09–7.65 (m, 3H), 7.56–7.29 (m, 3H), 3.79 (d, *J* = 3.0 Hz, 3H). ^13^C-NMR (101 MHz, CDCl_3_) δ 166.33 (s), 163.96 (s), 159.24 (s), 156.59 (s), 136.94 (s), 134.23 (s), 132.38 (s), 129.82 (s), 127.45 (s), 122.87 (s), 122.68 (s), 120.38 (s), 120.10 (s), 119.00 (s), 118.75 (s), 52.72 (s). HR-MS (ESI) [M + H]^+^ calcd for C_16_H_9_F_2_NO_4_: 318.0500, found: 318.0574.

Data for **3r** (2-(4-bromo-2,6-difluorophenyl)-4-fluoroisoindoline-1,3-dione): shite solid; yield, 78%; m.p.: 140–142 °C; ^1^H-NMR (400 MHz, CDCl_3_) δ 7.90–7.79 (m, 2H), 7.52 (t, *J* = 8.0 Hz, 1H), 7.32 (d, *J* = 6.9 Hz, 2H). ^13^C-NMR (101 MHz, CDCl_3_) δ 164.25 (s), 161.90 (s), 160.05 (s), 159.36 (s), 156.70 (s), 137.41 (s), 133.92 (s), 123.60 (s), 123.20 (s), 123.01 (s), 120.43 (s), 117.81 (s), 116.30 (s), 108.15 (s). HR-MS (ESI) [M + H]^+^ calcd for C_14_H_5_BrF_3_NO_2_: 355.9456, found: 355.9522.

Data for **3s** (4-fluoro-2-(4-isopropylphenyl)isoindoline-1,3-dione): shite solid; yield, 79%; m.p.: 116–117 °C; ^1^H-NMR (400 MHz, CDCl_3_) δ 7.68 (d, *J* = 2.4 Hz, 2H), 7.34 (ddd, *J* = 9.0, 6.1, 3.1 Hz, 1H), 7.25 (q, *J* = 8.6 Hz, 4H), 2.87 (dt, *J* = 13.8, 6.9 Hz, 1H), 1.19 (d, *J* = 6.9 Hz,6H).^13^C-NMR (101 MHz, CDCl_3_) δ 166.33 (s), 164.08 (s), 159.21 (s), 156.56 (s), 149.16 (s), 136.84 (s), 134.03 (s), 128.77 (s), 127.28 (s), 126.44 (s), 122.59 (s), 119.94 (s), 33.94 (s), 23.91 (s). HR-MS (ESI) [M + H]^+^ calcd for C_17_H_14_FNO_2_: 284.1009, found: 284.1086.

Data for **3t** (5-chloro-2-(3-fluoro-4-methylphenyl)isoindoline-1,3-dione): shite solid; yield, 79%; m.p.: 178–179 °C; ^1^H-NMR (400 MHz, CDCl_3_) δ 7.96–7.85 (m, 2H), 7.76 (dd, *J* = 8.0, 1.4 Hz, 1H), 7.31 (t, *J* = 8.1 Hz, 1H), 7.14 (d, *J* = 8.8 Hz, 2H), 2.33 (s, 3H). ^13^C-NMR (101 MHz, CDCl_3_) δ 166.07 (s), 165.76 (s), 162.14 (s), 141.29 (s), 134.61 (s), 133.29 (s), 131.68 (s), 129.67 (s), 125.10 (s), 124.26 (s), 121.79 (s), 113.65 (s), 113.40 (s), 99.99 (s), 14.41 (s). HR-MS (ESI) [M + H]^+^ calcd for C_15_H_9_ClFNO_2_: 290.0306, found: 290.0376.

Data for **3u** (5-chloro-2-(4-isopropylphenyl)isoindoline-1,3-dione): white solid; yield, 70%; m.p.: 137–138 °C; ^1^H-NMR (400 MHz, CDCl_3_) δ 7.90 (dd, *J* = 13.7, 4.7 Hz, 2H), 7.74 (dd, *J* = 8.0, 1.8 Hz, 1H), 7.34 (d, *J* = 8.6 Hz, 4H), 2.97 (dt, *J* = 13.8, 6.9 Hz, 1H), 1.28 (d, *J* = 6.9 Hz, 6H).^13^C-NMR (101 MHz, CDCl_3_) δ 166.50 (s), 166.19 (s), 149.16 (s), 141.06 (s), 134.41 (s), 133.50 (s), 129.88 (s), 128.91 (s), 127.29 (s), 126.34 (s), 124.97 (s), 124.14 (s), 33.94 (s), 23.91 (s). HR-MS (ESI) [M + H]^+^ calcd for C_17_H_14_ClNO_2_: 300.0713, found: 300.0791.

Data for **3v** (5-chloro-2-(1-methyl-1H-pyrazol-5-yl)isoindoline-1,3-dione): white solid; yield, 76%; m.p.: 175–176 °C; ^1^H-NMR (400 MHz, CDCl_3_) δ 7.95 (dd, *J* = 12.1, 4.7 Hz, 2H), 7.83 (dd, *J* = 8.0, 1.6 Hz, 1H), 7.61 (d, *J* = 1.9 Hz, 1H), 6.33 (d, *J* = 1.9 Hz, 1H), 3.78 (s, 3H).^13^C-NMR (101 MHz, CDCl_3_) δ 165.09 (s), 164.81 (s), 141.88 (s), 138.85 (s), 135.10 (s), 133.20 (s), 129.59 (s), 129.20 (s), 125.55 (s), 124.69 (s), 104.63 (s), 36.47 (s). HR-MS (ESI) [M + H]^+^ calcd for C_12_H_8_ClN_3_O_2_: 262.0305, found: 262.0384.

Data for **3w** (5-chloro-2-(thiophen-2-ylmethyl)isoindoline-1,3-dione): white solid; yield, 79%; m.p.: 93–94 °C; ^1^H-NMR (400 MHz, CDCl_3_) δ 7.89 (dd, *J* = 11.9, 4.9 Hz, 2H), 7.78 (d, *J* = 8.0 Hz, 1H), 7.39–7.22 (m, 2H), 7.04 (dd, *J* = 6.0, 2.4 Hz, 1H), 5.11 (s, 2H).^13^C-NMR (101 MHz, CDCl_3_) δ 166.60 (s), 166.27 (s), 140.81 (s), 137.67 (s), 134.12 (s), 133.72 (s), 130.11 (s), 127.90 (s), 126.94 (s), 126.05 (s), 124.70 (s), 123.93 (s), 35.91 (s). HR-MS (ESI) [M + H]^+^ calcd for C_13_H_8_ClNO_2_S: 277.9964, found: 278.0032.

Data for **3x** (2-(4-isopropylphenyl)-5-methylisoindoline-1,3-dione): white solid; yield, 47%; m.p.: 161–162 °C; ^1^H-NMR (400 MHz, CDCl_3_) δ 7.82 (d, *J* = 7.6 Hz, 1H), 7.75 (s, 1H), 7.57 (d, *J* = 7.5 Hz, 1H), 7.35 (s, 4H), 2.96 (dt, *J* = 13.7, 6.9 Hz, 1H), 2.54 (s, 3H), 1.28 (d, *J* = 6.9 Hz, 6H).^13^C-NMR (101 MHz, CDCl_3_) δ 167.66 (s), 167.54 (s), 148.77 (s), 145.67 (s), 134.89 (s), 132.22 (s), 129.32 (s), 129.25 (s), 127.19 (s), 126.44 (s), 124.21 (s), 123.62 (s), 33.94 (s), 23.94 (s), 22.07 (s). HR-MS (ESI) [M + H]^+^ calcd for C_18_H_17_NO_2_: 280.1259, found: 280.1338.

Data for **3y** (2-(4-isopropylphenyl)-5-nitroisoindoline-1,3-dione): white solid; yield, 57%; m.p.: 179–180 °C; ^1^H-NMR (400 MHz, CDCl_3_) δ 8.77 (d, *J* = 1.7 Hz, 1H), 8.67 (dd, *J* = 8.1, 1.9 Hz, 1H), 8.15 (d, *J* = 8.1 Hz, 1H), 7.44–7.30 (m, 4H), 3.09–2.84 (m, 1H), 1.29 (d, *J* = 6.9 Hz, 6H).^13^C-NMR (101 MHz, CDCl_3_) δ 165.11 (s), 151.97 (s), 149.66 (s), 136.21 (s), 133.21 (s), 129.56 (s), 128.51 (s), 127.44 (s), 126.23 (s), 124.97 (s), 119.14 (s), 33.97 (s), 23.90 (s). HR-MS (ESI) [M + H]^+^ calcd for C_17_H_14_N_2_O_4_: 311.0954, found: 311.1032.

### 3.4. Herbicidal Activity

The herbicidal activities of the test compounds against *B. campestris*, *A. retroflexus* and *D. sanguinalis* were evaluated by the small cup method and foliar spray method at 200 mg/L, according the following procedure [33,34]. With respect to the small cup method, generally, the test compound (20 mg) was dissolved in *N*,*N*-dimethylformamide (DMF) (1 mL) and then diluted in water containing 0.1% Tween 80 to a final concentration of 200 mg/L. The controls were treated with the same solution but without any test compound. A piece of filter paper in a 50 mL beaker was treated with the test compound solution (1 mL), and then 10 seeds that were soaked in water for 24 h were added. All treatments were repeated three times.

The pre- and post-emergence herbicidal activities of the title compounds were evaluated at an application rate of 90 g ai/ha in a greenhouse according to a reported method [35,36]. Nine seeds of these plants (*A. retroflexus* seeds, *B. campestris* seeds or *D. sanguinalis* seeds) were sown at a depth of 5 mm in a mixture of vermiculite/nutrient-enriched soil (1:1, m/m) with some water at 4 cm below the surface and then cultivated at 20–25 °C. To test the pre-emergence herbicidal activities, the abovementioned soil was sprayed with the title compound solution before germination, and the results were determined after two weeks. To test the post-emergence herbicidal activities, plants at the 2–4 leaf stage after germination were treated with the test compound solution by a walking spray tower and then cultivated for one week. The percentage of herbicidal activity was calculated by comparing the fresh weight of the growth-inhibited plants with that of the healthy control plants, where completely inhibited growth was set as 100 and the healthy control was set as 0.

### 3.5. PPO Enzyme Assays

To further explore the mode of action of these target compounds, **3a** was selected as a representative to confirm whether it can act on PPO. Briefly, the procedures were followed as described here. *D. sanguinalis* plants were treated with 90 g ai/ha of compound **3a** and Flumioxazin by the post-emergence method. After 3 days, *D. sanguinalis* leaves (0.2 g) were collected and dissolved in extraction medium (1.5 mL) in an ice bath, and the mixture was then centrifuged at 4 °C × 12,000 rpm for 15 min. A polyphenol oxidase kit (G0113W, Suzhou Grace Bio-technology Co., Ltd., Suzhou, China) was obtained from commercial sources to determine the PPO activity. The change in absorbance (every 5 min) was measured on a POLARstar Optima/Galaxy instrument (BMG) (Shanghai Microplate Co., Ltd., Shanghai, China) at 420 nm [37,38,39]. One unit of PPO activity was defined as a change in absorbance of 0.01 per minute.

## 4. Conclusions

In summary, a series of phthalimide derivatives were designed by molecular docking and by using Flumioxazin as a lead, then synthesized and characterized by NMR, HR-MS, and the typical crystal structure was determined by X-ray diffraction. The herbicidal activities of these compounds were assessed against *B. campestris*, *A. retroflexus* and *D. sanguinalis*, by the small cup, pre-emergence, and post-emergence methods, respectively. Most of the synthesized compounds exhibited good to excellent herbicidal activities, and especially **3a** displayed the same efficacy against *A. retroflexus* and *B. campestris* to commercial standards of Flumioxazin. Further PPO activity assays confirmed that the mode of action of **3a** is similar to PPO inhibitors. These results suggest that our molecular design strategy is effective.

## Figures and Tables

**Figure 1 molecules-24-04363-f001:**
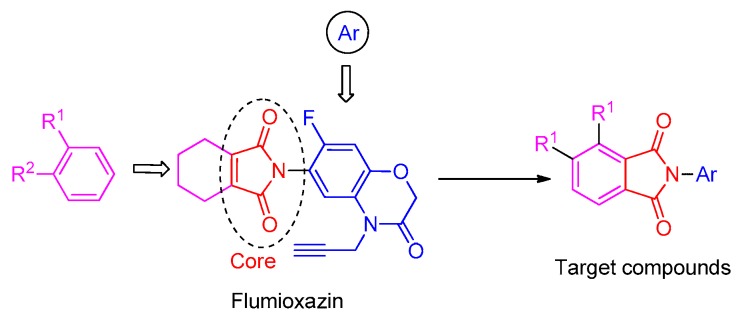
Design of the title compounds.

**Figure 2 molecules-24-04363-f002:**
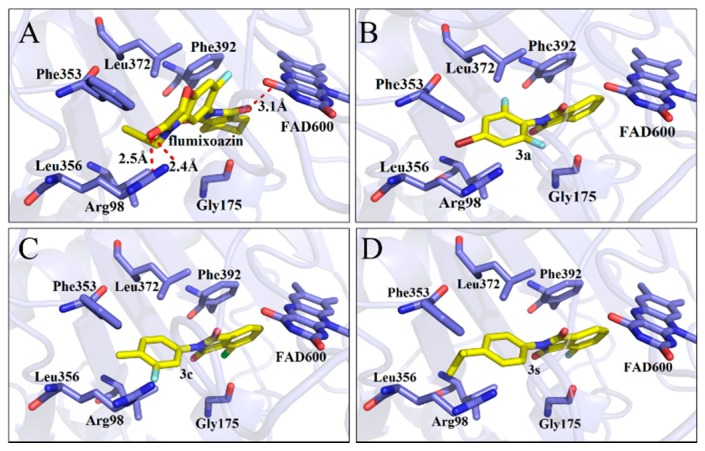
Docking model shown in Pymol, molecule (blade yellowing) and residues of *mt*PPO (wrinkling); Flumioxazin (**A**), **3a** (**B**), **3c** (**C**), and **3s** (**D**).

**Figure 3 molecules-24-04363-f003:**
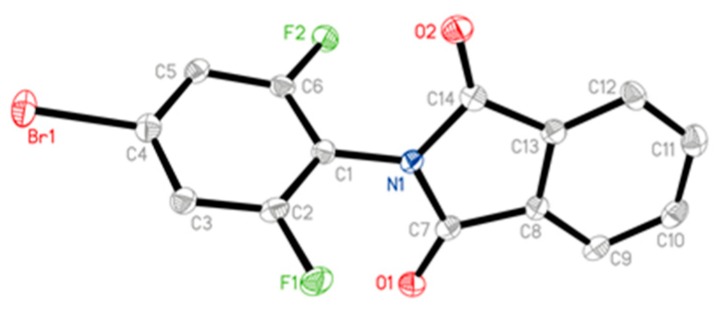
Crystal structure for **3a** by X-ray diffraction determination.

**Figure 4 molecules-24-04363-f004:**
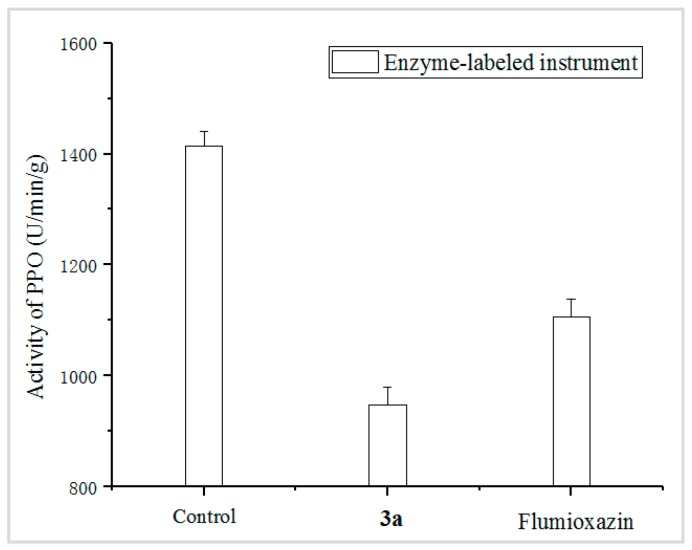
PPO enzyme activity involved by **3a** and Flumioxazin. (The control was sprayed with blank solution without any compounds; the **3a** and Flumioxazin were sprayed with **3a** and Flumioxazin at 90 g ai/ha, respectively).

**Table 1 molecules-24-04363-t001:** The affinity between compounds and *mt*PPO. (Affinity, Kcal/mol).

Compd.	Affinity	Compd.	Affinity	Compd.	Affinity	Compd.	Affinity
**3a**	−10.0	**3h**	−8.7	**3o**	−9.5	**3v**	−8.1
**3b**	−8.6	**3i**	−8.4	**3p**	−8.7	**3w**	−7.7
**3c**	−10.1	**3j**	−7.7	**3q**	−9.3	**3x**	−9.8
**3d**	−9.1	**3k**	−9.4	**3r**	−9.0	**3y**	−9.5
**3e**	−9.8	**3l**	−8.9	**3s**	−10.2	Flumioxazin	−9.5
**3f**	−9.7	**3m**	−9.9	**3t**	−9.7	Chlortoluron	−7.1

**Table 2 molecules-24-04363-t002:** General Synthetic Route for Compounds **3a–****3y.**

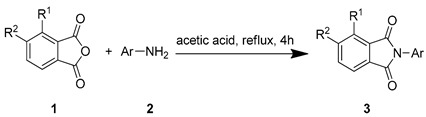														
Compd.	R^1^	R^2^	Ar	Yield (%)	Compd.	R^1^	R^2^	Ar	Yield (%)	Compd.	R^1^	R^2^	Ar	Yield (%)
**3a**	H	H	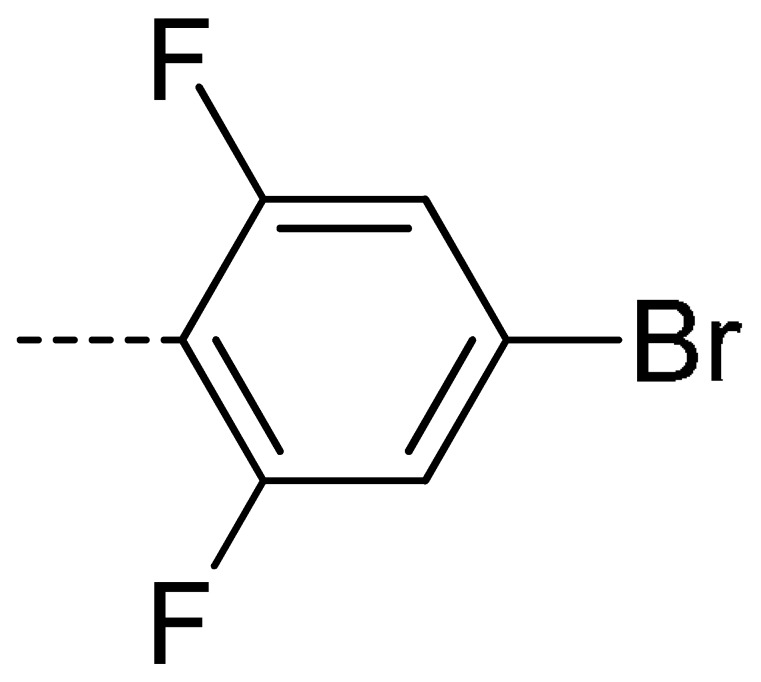	73	**3j**	Cl	H	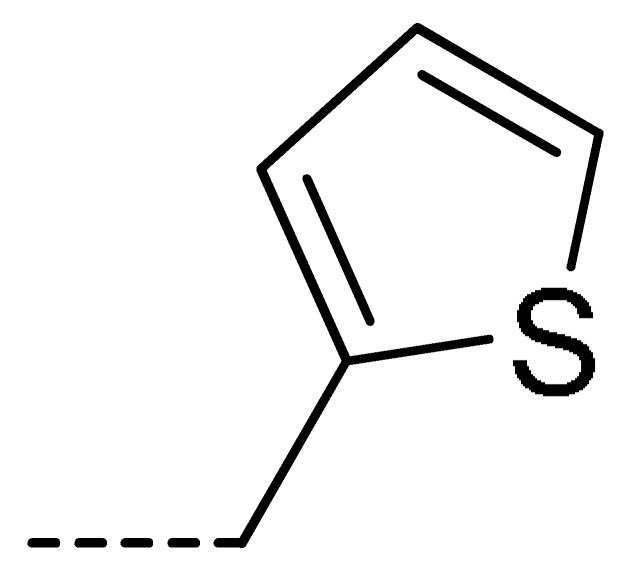	71	**3s**	F	H	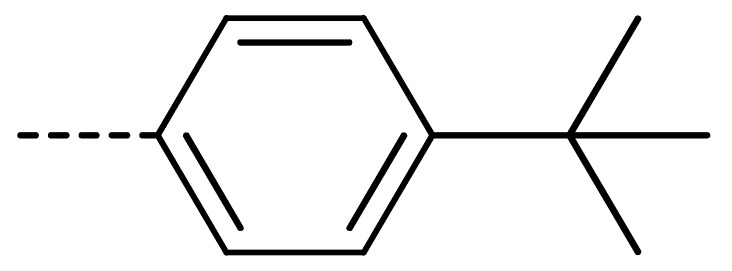	79
**3b**	H	Cl	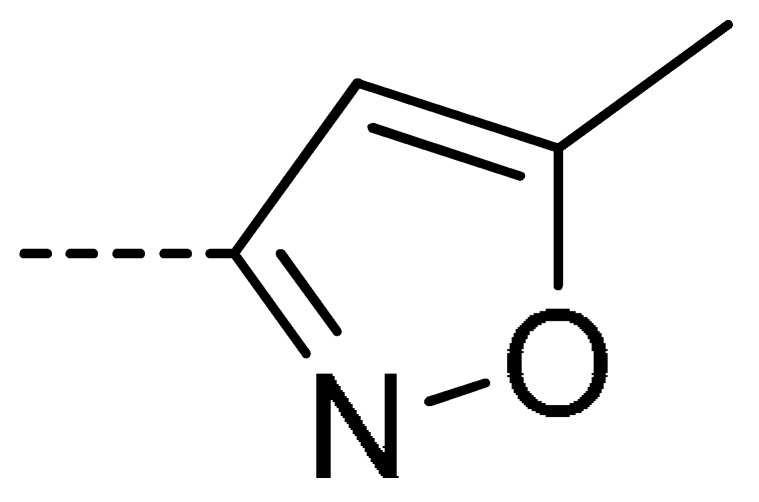	77	**3k**	Cl	H	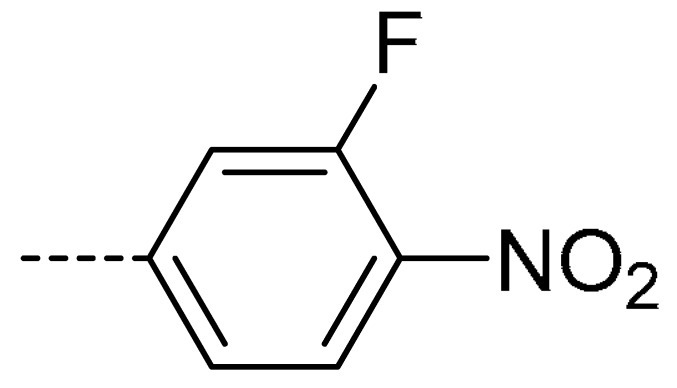	63	**3t**	H	Cl	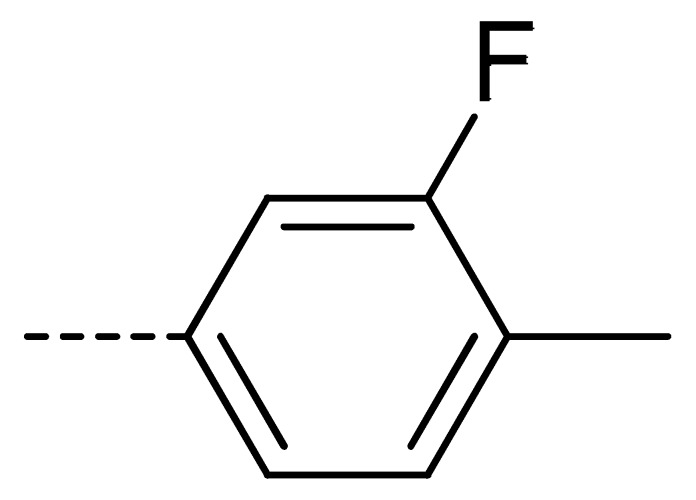	79
**3c**	Cl	H	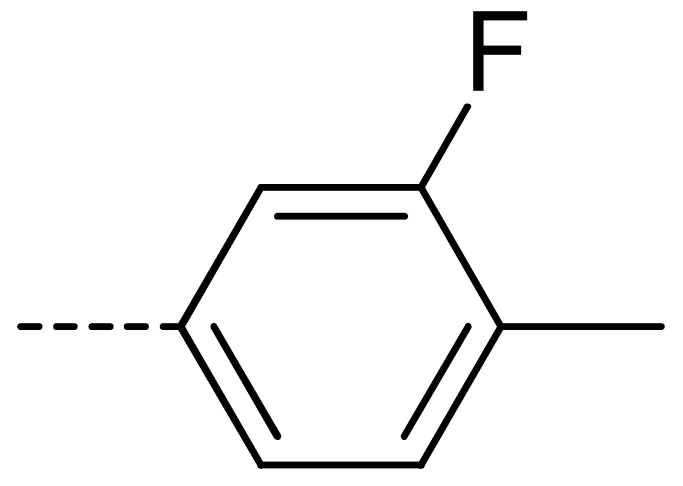	81	**3l**	Cl	H	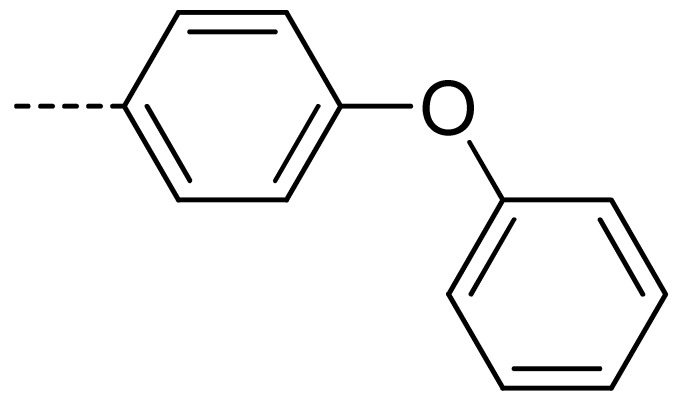	86	**3u**	H	Cl	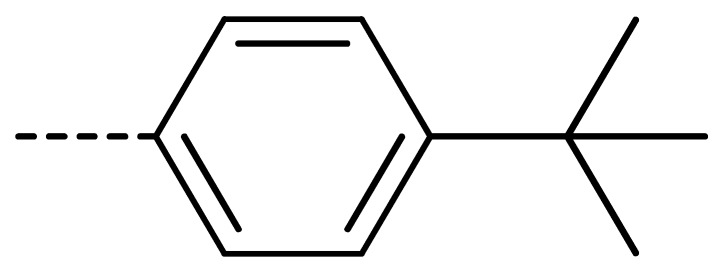	70
**3d**	Cl	H	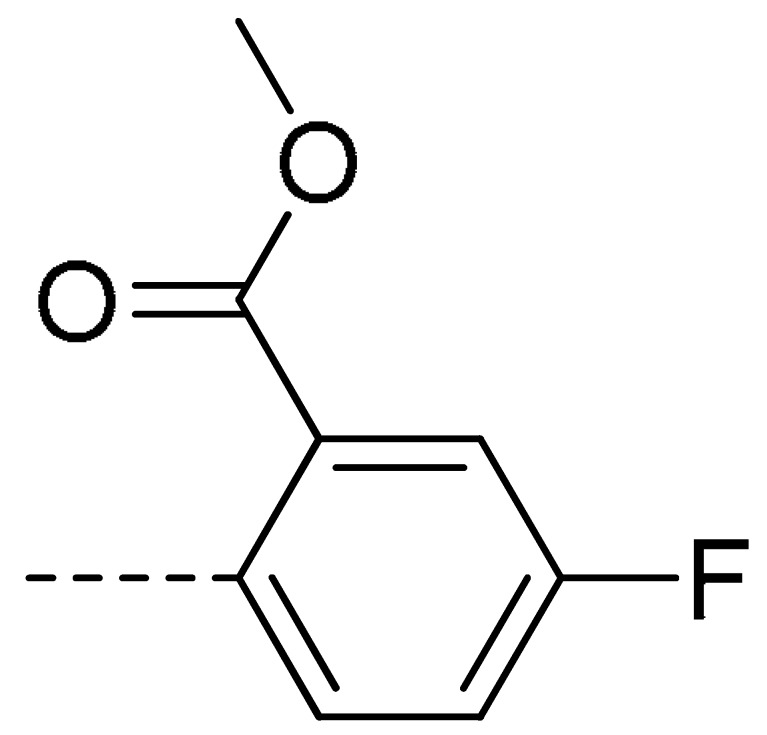	28	**3m**	Cl	H	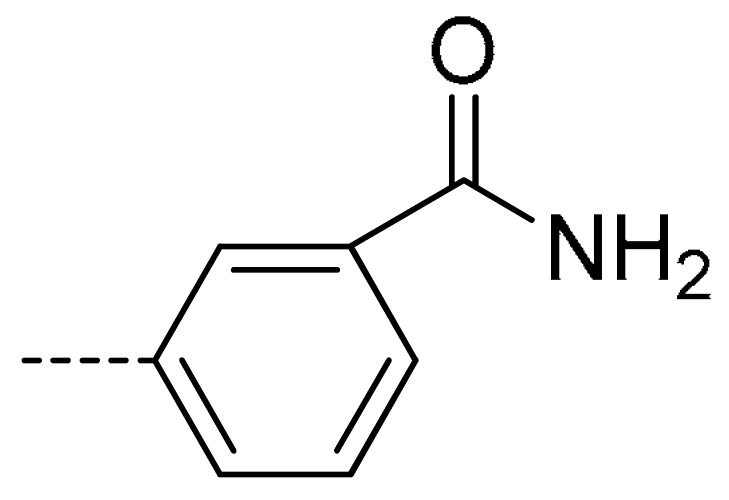	84	**3v**	H	Cl	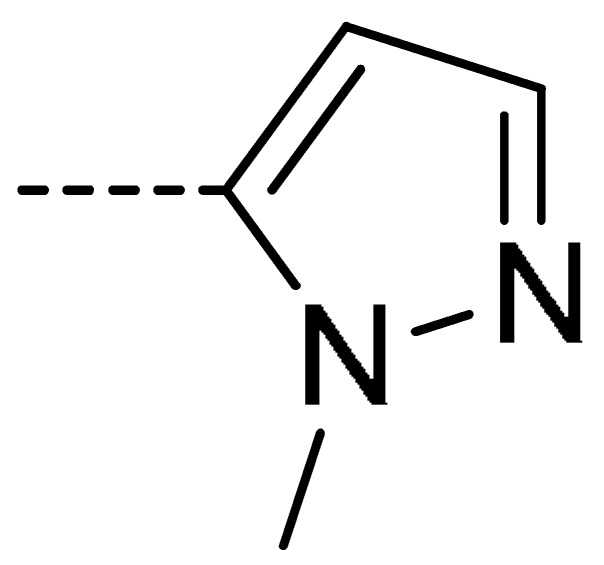	76
**3e**	Cl	H	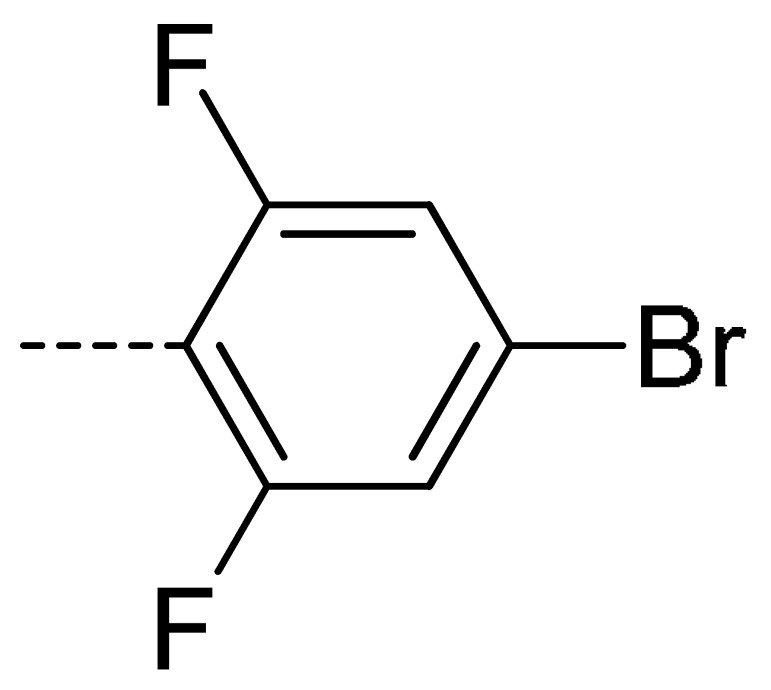	80	**3n**	Cl	H	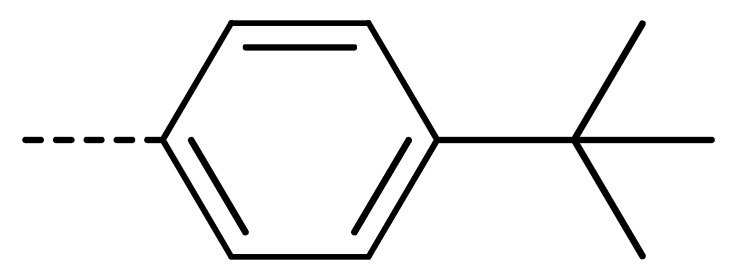	52	**3w**	H	Cl	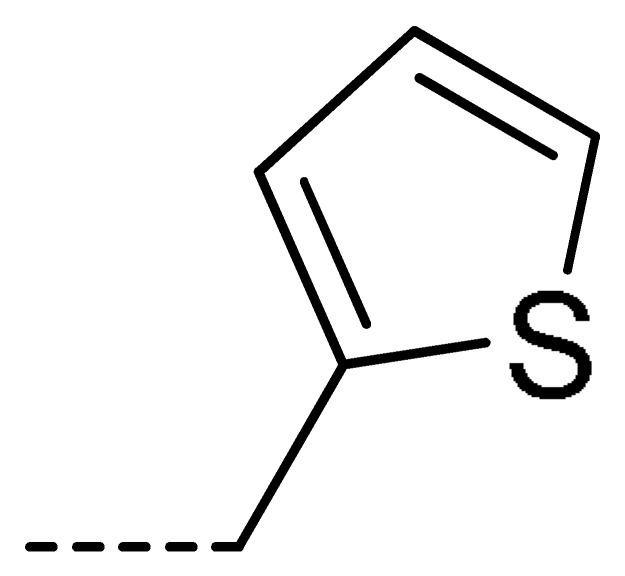	79
**3f**	Cl	H	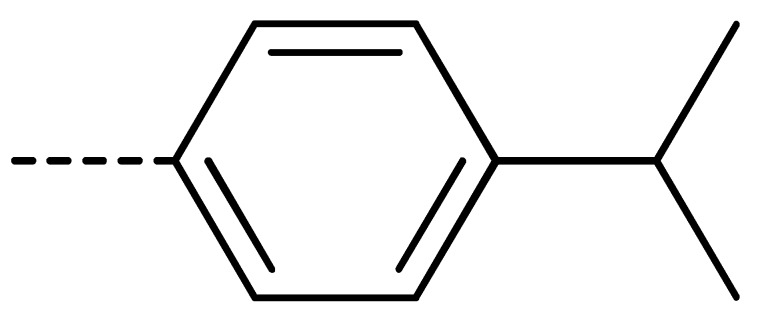	78	**3o**	Cl	H	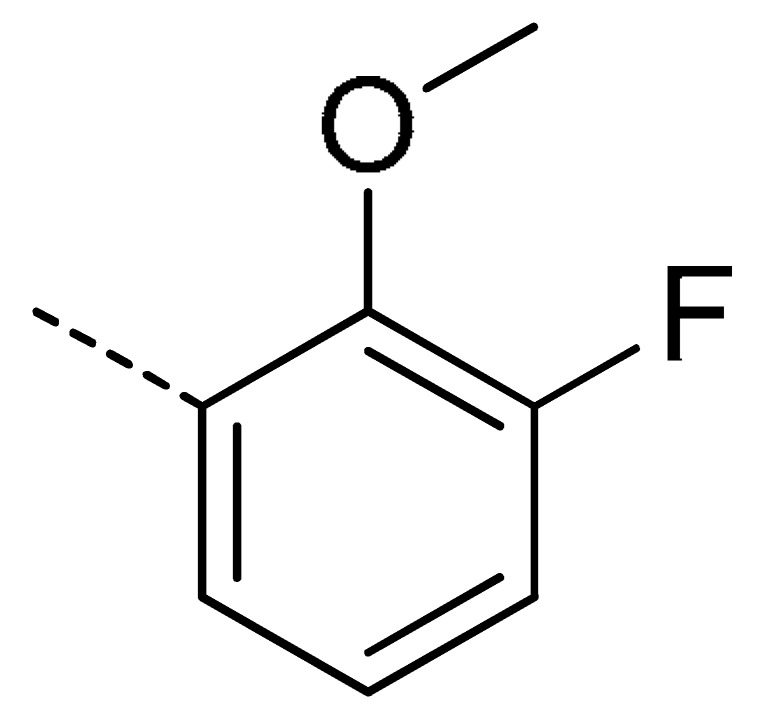	87	**3x**	H	CH_3_	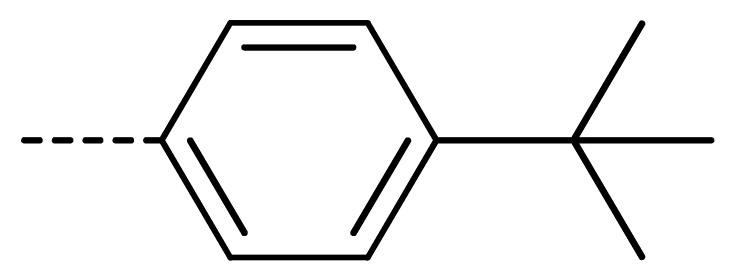	47
**3g**	Cl	H	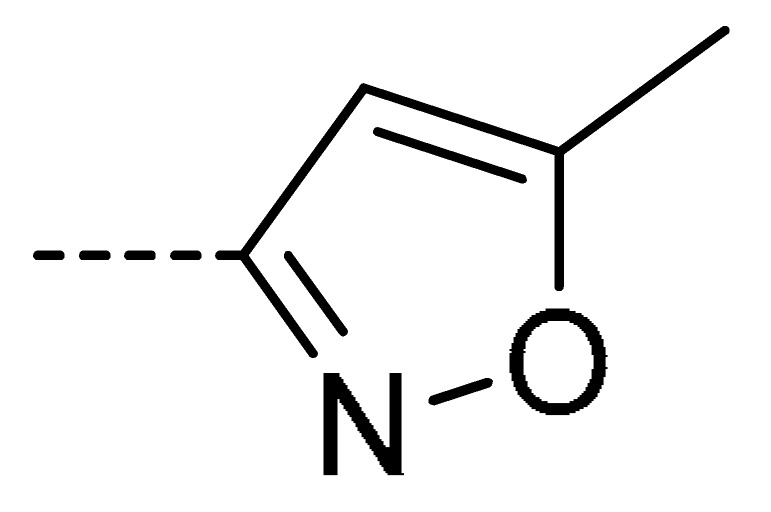	58	**3p**	Cl	H	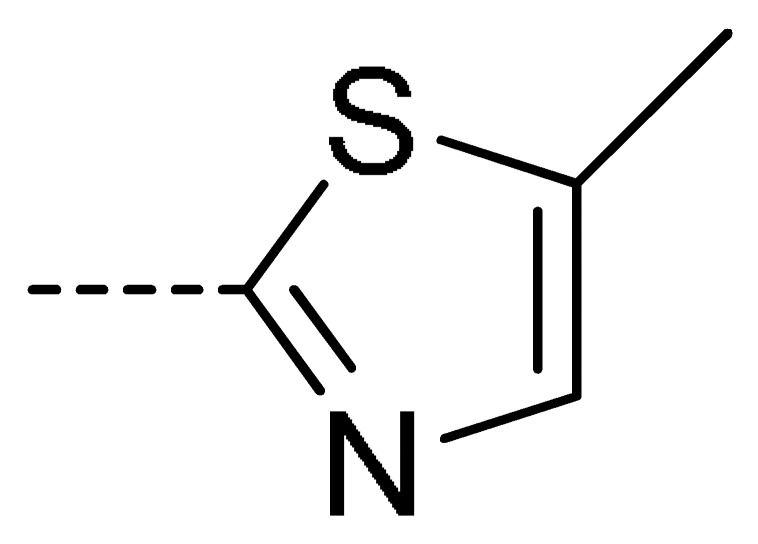	30	**3y**	H	NO_2_	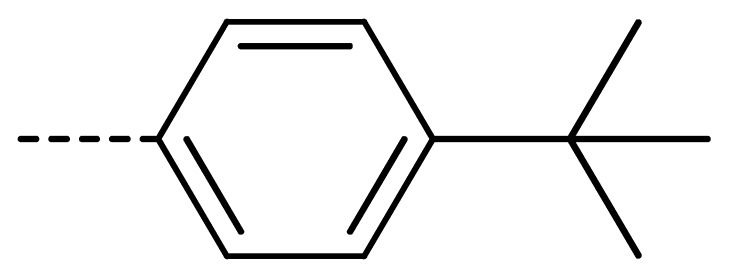	57
**3h**	Cl	H	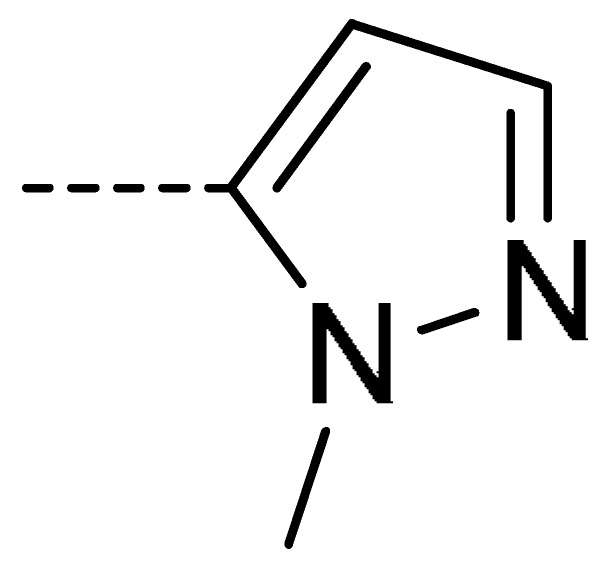	60	**3q**	F	H	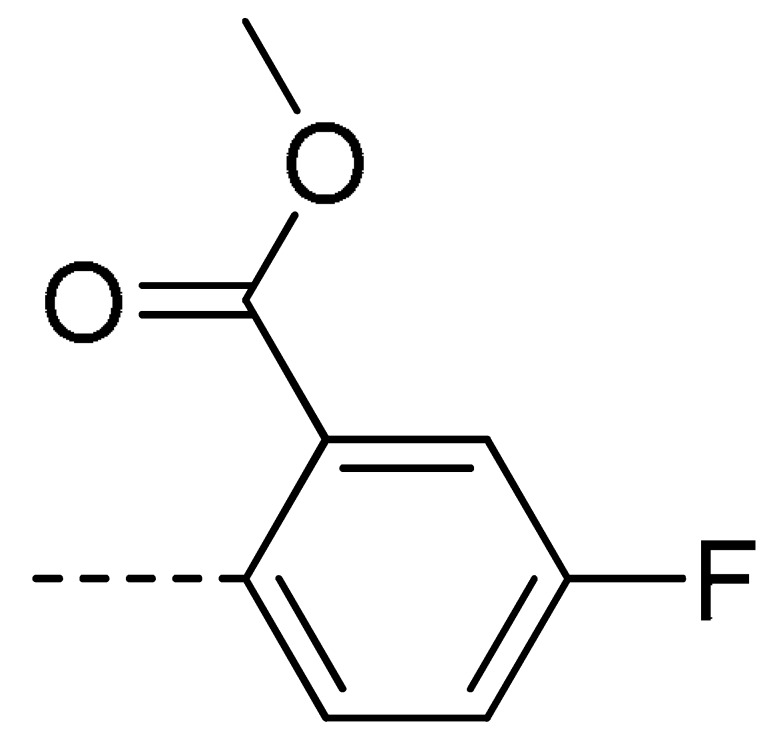	57					
**3i**	Cl	H	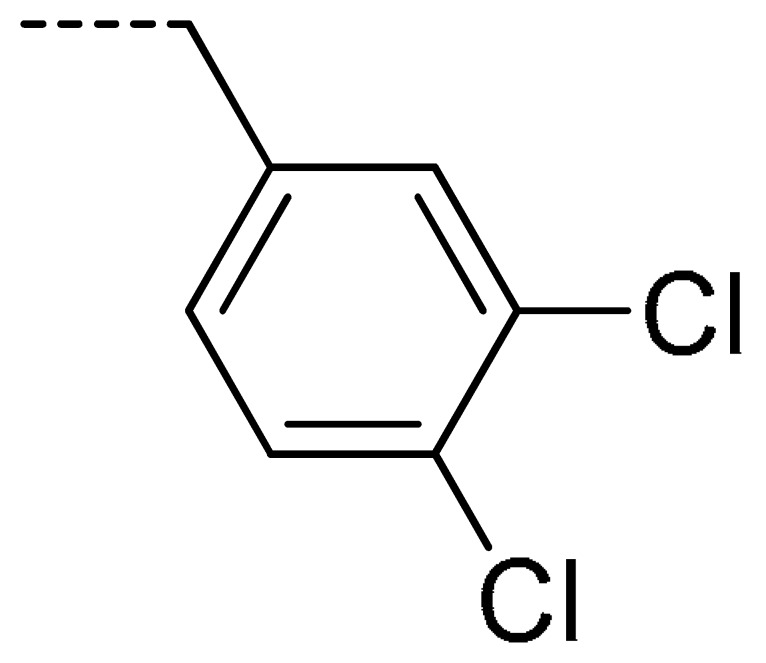	67	**3r**	F	H	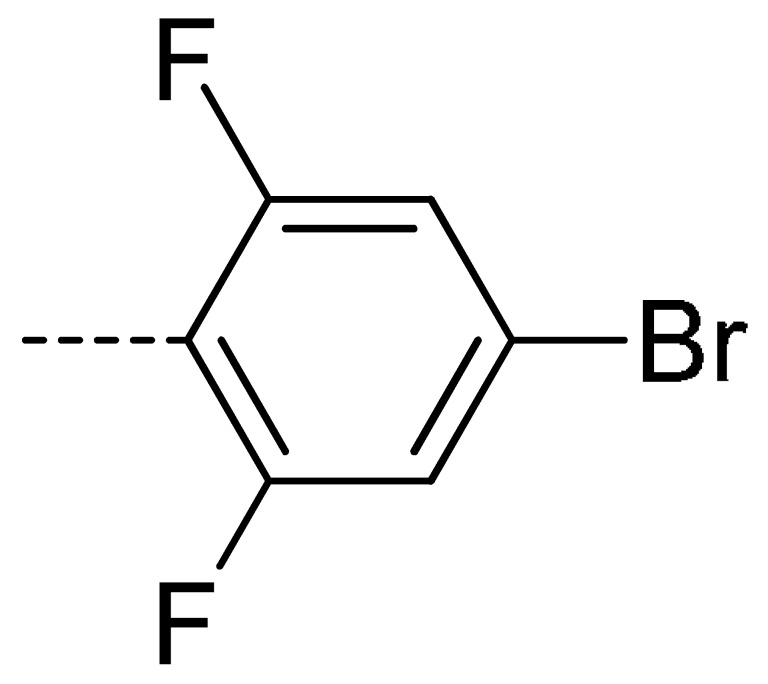	78					

**Table 3 molecules-24-04363-t003:** Herbicidal activity of compounds **3a**–**3y** at 200 mg/L by the small cup method.

Compd.	*BC* ^a^		*AR* ^a^		*DS* ^a^	
	Root	Stem	Root	Stem	Root	Stem
**3a**	92 ± 1	61 ± 2	37 ± 1	87 ± 2	68 ± 3	83 ± 2
**3b**	74 ± 2	0	28 ± 4	66 ± 4	77 ± 2	62 ± 3
**3c**	78 ± 5	0	65 ± 1	22 ± 1	61 ± 1	0
**3d**	82 ± 1	35 ± 1	81 ± 4	33 ± 2	65 ± 2	10 ± 1
**3e**	28 ± 2	21 ± 2	51 ± 1	55 ± 3	28 ± 2	38 ± 3
**3f**	46 ± 2	73 ± 3	68 ± 0	42 ± 0	47 ± 3	0
**3g**	0	18 ± 0	74 ± 2	41 ± 2	91 ± 1	83 ± 1
**3h**	77 ± 3	0	62 ± 2	33 ± 1	55 ± 1	34 ± 1
**3i**	58 ± 4	0	55 ± 1	27 ± 2	67 ± 5	17 ± 3
**3j**	75 ± 0	0	66 ± 1	34 ± 1	87 ± 2	11 ± 0
**3k**	65 ± 1	0	53 ± 3	36 ± 2	57 ± 1	0
**3l**	68 ± 2	6 ± 2	44 ± 3	31 ± 2	48 ± 4	51 ± 3
**3m**	45 ± 3	0	55 ± 4	45 ± 3	57 ± 2	0
**3n**	49 ± 1	0	59 ± 2	33 ± 2	59 ± 4	0
**3o**	89 ± 1	77 ± 0	83 ± 5	51 ± 3	63 ± 4	0
**3p**	51 ± 2	9 ± 2	37 ± 3	72 ± 4	58 ± 1	45 ± 2
**3q**	28 ± 1	22 ± 1	22 ± 1	21 ± 2	38 ± 3	0
**3r**	65 ± 4	54 ± 3	62 ± 2	75 ± 3	87 ± 2	86 ± 2
**3s**	64 ± 2	15 ± 1	35 ± 4	61 ± 5	53 ± 3	28 ± 5
**3t**	54 ± 1	0	16 ± 1	39 ± 4	46 ± 2	18 ± 1
**3u**	77 ± 1	53 ± 3	0	53 ± 2	51 ± 2	32 ± 1
**3v**	60 ± 5	0	10 ± 2	42 ± 3	22 ± 1	21 ± 2
**3w**	58 ± 2	0	28 ± 1	61 ± 2	79 ± 2	11 ± 2
**3x**	59 ± 4	0	36 ± 3	55 ± 3	31 ± 2	24 ± 1
**3y**	68 ± 2	9 ± 0	34 ± 4	60 ± 4	44 ± 1	37 ± 3
Chlortoluron	85 ± 4	58 ± 3	92 ± 3	90 ± 5	98 ± 0	97 ± 1
Atrazine	81 ± 1	52 ± 1	32 ± 2	66 ± 2	58 ± 1	60 ± 2
Flumioxazin	85 ± 3	72 ± 5	82 ± 2	88 ± 1	71 ± 2	91 ± 0

^a^*BC* for *B. campestris*; *AR* for *A. retroflexus*; *DS* for *D. sanguinalis*.

**Table 4 molecules-24-04363-t004:** Post-emergence herbicidal activity of compounds **3a**–**3y** at 90 g ai/ha.

Compd.	*BC*	*AR*	*DS*	Compd.	*BC*	*AR*	*DS*
**3a**	25 ± 2	82 ± 3	37 ± 1	**3o**	16 ± 2	27 ± 2	23 ± 2
**3b**	27 ± 2	18 ± 2	12 ± 1	**3p**	19 ± 4	20 ± 1	14 ± 0
**3c**	48 ± 1	13 ± 1	19 ± 3	**3q**	30 ± 2	19 ± 2	36 ± 1
**3d**	60 ± 3	73 ± 2	20 ± 1	**3r**	37 ± 1	22 ± 3	36 ± 5
**3e**	35 ± 4	47 ± 3	33 ± 1	**3s**	22 ± 1	44 ± 2	18 ± 1
**3f**	56 ± 1	10 ± 2	11 ± 1	**3t**	29 ± 2	62 ± 3	17 ± 1
**3g**	38 ± 2	0	23 ± 2	**3u**	60 ± 2	5 ± 1	28 ± 2
**3h**	58 ± 1	7 ± 1	27 ± 2	**3v**	33 ± 1	0	0
**3i**	28 ± 2	34 ± 2	37 ± 1	**3w**	35 ± 1	22 ± 4	36 ± 4
**3j**	23 ± 2	20 ± 5	31 ± 1	**3x**	17 ± 2	7 ± 2	7 ± 1
**3k**	35 ± 5	24 ± 3	0	**3y**	28 ± 1	0	0
**3l**	33 ± 1	37 ± 2	26 ± 2	Chlortoluron	nd ^a^	91 ± 1	65 ± 2
**3m**	22 ± 4	35 ± 1	34 ± 2	Atrazine	nd ^a^	91 ± 2	47 ± 3
**3n**	25 ± 2	30 ± 5	11 ± 1	Flumioxazin	85 ± 6	92 ± 4	86 ± 6

^a^ nd, not detect.

**Table 5 molecules-24-04363-t005:** Per-emergence herbicidal activity of compounds **3a**, **3d** and Flumioxazin at 90 g ai/ha.

Compd.	*AR*	*DS*
**3a**	98 ± 2	61 ± 2
**3d**	36 ± 3	0
Flumioxazin	100	100

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
