# Peer review of "Design and Synthesis of N-phenyl Phthalimides as Potent Protoporphyrinogen Oxidase Inhibitors"

_molecules, 2019, doi:10.3390/molecules24234363_

Round 1

Reviewer 1 Report

The authors answered my questions and allegations very well.

The authors also provided relevant supplementary materials that I asked about.

In the introduction, I recommend the authors cite the following papers:

Karcz D. et al. Spectroscopic and theoretical investigation into substituent- and aggregation-related dual fluorescence effects in the selected 2-amino-1,3,4-thiadiazoles, Journal of Molecular Liquids
Starzak K. et al. Fluorescence quenching-based mechanism for determination of hypochlorite by coumarin-derived sensors, International Journal of Molecular Sciences
Niemczynowicz A. et al. Spectroscopic and Theoretical Studies of Dual Fluorescence in 2-Hydroxy-N-(2-phenylethyl)benzamide Induced by ESIPT Process – Solvent Effect, Journal of Luminescence
Czernel, G. et al. Spectroscopic Studies of Dual Fluorescence in 2-(4-Fluorophenylamino)-5-(2, 4-dihydroxybenzeno)-1, 3, 4-thiadiazole: Effect of Molecular Aggregation in a Micellar System. Molecules

No future review is necessary.

Author Response

Thank you for your suggestion, we have read the above references but they were not related to our research content, so we couldn’t cite them.

Reviewer 2 Report

After the revision, the manuscript is still NOT ready for publishing. The author should read some papers on Molecules or other journals to learn how to organize and present their data. The following items should also be addressed prior to publication:

In the Abstract, for compound 3a, 3d, 3g, 3j, and 3r, giving the substituents of R1, R2… without the structure is meaningless. Please give the names of 3a, 3d, 3g, 3j, and 3r (or describe the functional groups modified on the phenyl phthalimide core structure). For example, "bromodifluorophenyl phthalimide 3a" can be used for 3a.      The author is still confused about what to keep at the Results and Discussions and what to keep at Experimental Section. Please read some papers published in Molecules or JOC on organic synthesis.           Please include the yields of 3a-3y in the table 2                                   Some of the final compounds 3a-3y were published before, thus include the references so that the author would know what compounds are novel and what were reported.

Author Response

This manuscript is a resubmission of an earlier submission. The following is a list of the peer review reports and author responses from that submission.

Round 1

Reviewer 1 Report

Recommendation: Publish after minor revisions noted.

Comments:

This manuscript by Gao reports the synthesis of N-phenyl phthalimides and the characterization of them as PPO inhibitors for their applications as herbicides. Some of the phthalimide derivatives, especially compounds 3a, 2-(4-bromo-2,6-difluorophenyl)isoindoline-1,3-dione, showed excellent herbicidal activity. This work appears to have not been reported before. The following items should be addressed prior to publication:

In the Abstract, please give the names of 3a, 3d, 3g, 3j, and 3r (or describe the functional groups modified on the phenyl phthalimide core structure). Otherwise, numbers do NOT give any information to the audiences when they are reading the abstract. In the Abstract, please provide full name of mtPPO, mitochondrial PPO, when it is shown in the manuscript for the first time. Line 30, please provide references for pyraclonil,Ref carfentrazone, Ref and fomesafen. Ref Line 30-31, what do you mean here by "Additional ones will be released in future"? If you know something new is going to be released, please include references. Otherwise, just delete this sentence. Even though the synthesis of N-phenyl phthalimides using acetic acid as solvent under reflux is a easy synthetic strategy, please still include a paragraph describing the synthesis. You can include something like how the substituents on the phenyl amines affect the reaction. Please include the yields of 3a-3y in Table 1 Put a space between acetic acid in the reaction condition at the scheme in Table 1. Table 1 including yields should belong to Results and Discussion. Line 90, differences activity = activity differences? In Figure 3, CK means control? Include the concentrations of 3a and flumioxazin at the paragraph of 3. PPO enzyme assays and in the Figure 3. Please calculate and show the % of inhibition of the PPO activity. Capital compounds 3 at line 120 (compounds 3) The results from at line 126-131 should be moved to Results and Discussion. Table 5 can be moved to Supporting Information.

Reviewer 2 Report

The submitted manuscript attempts reporting on the design and synthesis of a series of phthalimide-derived protoporphyrinogen oxidase inhibitors. Generally the idea of synthesis based on the molecular docking studies is quite interesting, however the second part of the manuscript namely the synthesis rises serious doubts due to a number of shortcomings and inconsistencies.

The authors state that the starting materials were commercially available, but the actual source (company name and purity grade of the compounds used) are not specified. These information, together with grade and the origin of the remaining chemicals used should be given in materials and methods section. The authors state that “the chemical structures of the target compounds 3 were confirmed by 1H-NMR, 13C-NMR and HRMS (see supporting data)” but there is no supporting data file available making the verification of the results obtained by authors impossible. Regardless the lack of NMR and MS data, the structural characterisation and the discussion of the novel structures based on only one crystal structure is insufficient. At least one representative example of the NMR spectrum should be given as a figure in the main text together with an extensive discussion of the NMR spectral features. The novel compounds should also be characterised with other spectroscopic techniques such as FT-IR , and the relevant data should be presented. The authors state that the melting points were determined but these data are also missing An elemental analysis (C, H, N) should be performed for all the novel compounds and the appropriate table should be added into the main text or into the supplementary data. The manuscript requires a professional grammar and language editing.

These issues have to be addressed in the thoroughly revised version of this manuscript. In its current form the manuscript is not suitable for publication in Molecules. Therefore I recommend rejecting this manuscript with the optional resubmission possibility in the future.

Reviewer 3 Report

Line 28: has been synthesized

Line 36: an N-substituted..

Table 1 should belong to the Results and Discussion part of the article, not to the Introduction.

In the scheme of Table 1..there should be a space between “acetic” and “acid”.

In Figure 2, you present the docking results for compounds 3s (B), 3a (C), 3c (D). It would be preferable to present the results for the compounds ordered alphabetically, ex: 3a, 3b and 3s.

The manuscript is not at all well organized and the presentation of the data is confusing. The title mentioned “synthesis”, but there is no information on this in the Results and discussion part. After the docking analysis, it is mandatory to discuss the chemical synthesis in chapter 2.2 and only after, the herbicidal activity. Otherwise, in the herbicidal activity, you present the results obtained for some of the compounds, but is not at all clear…these compounds have been synthesized previously or now, in the present work. So, it is absolutely necessary to insert the chemical synthesis with discussions before the evaluation of the activity.